# The constraint of CO<sub>2</sub> measurements made onboard passenger aircraft on surfaceatmosphere fluxes: the impact of transport model errors in vertical mixing

Shreeya Verma<sup>1</sup>, Julia Marshall<sup>1</sup>, Christoph Gerbig<sup>1</sup>, Christian Roedenbeck<sup>1</sup>, Kai Uwe Totsche<sup>2</sup>

<sup>1</sup>Max Planck Institute for Biogeochemistry, Jena, 07745, Germany. <sup>2</sup>Institute of Geosciences, Friedrich Schiller University, Jena, 07749, Germany. Correspondence to: S. Verma (sverma@bgc-jena.mpg.de)

Abstract. Inaccurate representation of atmospheric processes by transport models is a dominant source of uncertainty in inverse analyses and can lead to large discrepancies in the retrieved flux estimates. We investigate the impact of

- 10 uncertainties in vertical transport as simulated by atmospheric transport models on fluxes retrieved using vertical profiles from aircraft as an observational constraint. Our numerical experiments are based on synthetic data with realistic spatial and temporal sampling of aircraft measurements. The impact of such uncertainties on the flux retrieved using the ground-based network with those retrieved using the aircraft profiles are compared. We find that the posterior flux retrieved using aircraft profiles is less susceptible to errors in boundary layer height as compared to the ground-based network. This finding
- 15 highlights a benefit of utilizing atmospheric observations made onboard aircraft over surface measurements for flux estimation using inverse methods. We further use synthetic vertical profiles of  $CO_2$  in an inversion to estimate the potential of these measurements, which will be made available through the IAGOS (In-Service Aircraft for a Global Observing System) project in future, in constraining the regional carbon budget. Our results show that the regions tropical Africa and temperate Eurasia, that are under-constrained by the existing surface based network, will benefit the most from these measurements, with the reduction of posterior flux uncertainty of about 7 to 10 %.


## **1** Introduction

Reliable prediction of climate change scenarios requires a thorough understanding the carbon-climate feedbacks in the earth system, and accurately estimating current sources and sinks of carbon is of prime importance. While it is impossible to measure these sources and sinks directly everywhere around the globe, we may estimate these using the 'top-down' 25 approach employing atmospheric observations in combination with knowledge of atmospheric transport and prior knowledge of the fluxes by inverse modelling. The inverse modelling scheme exploits the fact that the spatial and temporal variations of atmospheric trace gases like CO<sub>2</sub> contain information about the exchange processes between the atmosphere and the surface of the earth. Unfortunately, the estimates of surface fluxes using this approach are prone to large uncertainties that can largely be attributed to imperfections in the transport models and insufficient data coverage by the observation network (Gerbig et al., 2003).


Atmospheric transport models use meteorological input like wind fields to link the observed atmospheric concentrations of tracers to the estimated fluxes at the surface of the earth. These models are not able to perfectly simulate atmospheric transport processes, which results in uncertainties in the retrieved surface fluxes (Law et al., 1996, 2008; Gerbig et al., 2003; 35 Stephens et al., 2007; Lauvaux et al., 2009; Houweling et al., 2010). One of the dominant sources of transport model uncertainty is the inaccurate representation of the vertical mixing near the surface of the earth and hence the boundary layer height (Stephens et al., 2007; Gerbig et al., 2008). An accurate simulation of the vertical mixing in the boundary layer accurately is critical since it is this part of the atmosphere where most observations are made and that lies closest to the

modelled tracer mixing ratios as well as the retrieved fluxes (Denning et al. 1996, 2008; Yi et al. 2004; Ahmadov et al.

carbon sources and sinks. Hence, misrepresentation of transport in the boundary layer can lead to significant biases in

2009).

Furthermore, a weak observational constraint due to insufficient atmospheric data is also an important factor that causes large errors in retrieved fluxes. Lack of measurements in the atmosphere or an unevenly distributed network of observation sites can result in a poorly constrained regional carbon budget (Gurney et al. 2002). Hence in addition to improved transport models, an enhanced global network of atmospheric measurements is indispensable for more accurate and precise estimation of surface fluxes using inverse modelling.

The current global measurement network of greenhouse gases combines in-situ measurements made by the ground-based stations and satellite instruments measuring total column mixing ratios remotely. While ground-based measurements are highly precise, the main limitation of these measurements is the sparse and uneven spatial coverage (Bousquet et al., 2006;

- Marquis and Tans, 2008). While parts of Europe and North America dispose of a fairly high data coverage from the surfacebased observation network, the tropical regions of Amazonia, Africa, remote regions of tundra, and Siberia are not adequately covered, sometimes even lacking measurements entirely. In addition, these measurements except those obtained from tall towers, are often not representative of large areas and provide information only at the local scale (Haszpra et al. 1999). Satellites largely overcome this drawback of ground-based measurements since they have the ability to provide
- information around the world using a single instrument. However, they have their limitations, too, which limits their use for accurate flux estimation using inverse methods. Space borne measurements are still somewhat limited by higher measurement uncertainty and systematic errors, as well as temporal heterogeneity in their sampling (Ehret and Kiemle, 2005; Galli et al. 2014; Checa-Garcia et al. 2015)
- The use of passenger aircraft as platforms for obtaining information about the physical and state and chemical composition
  of the atmosphere is a rather new concept. IAGOS (In-Service Aircraft for a Global Observing System) is a European Research Infrastructure that deploys sensors on commercial airliners that make regular in-situ measurements of the atmosphere. The project is an extension and continuation of the MOZAIC (Measurement of Ozone and Water Vapour on Airbus in-service Aircraft) project (Marenco et al., 1998) that was initiated in the year 1993. Detailed and continuous measurements are made during long distance flights by on board instruments, thus providing a view of the horizontal and vertical distribution of the measured trace gases at high temporal and spatial resolution. The last MOZAIC aircraft was deactivated in October 2014; currently six IAGOS aircraft are flying. IAGOS provides observations with applications in the field of atmospheric modelling and for validation of satellite observations. There are a number of species that are currently being measured by IAGOS aircraft like CO, O<sub>3</sub>, NO<sub>x</sub>, NO<sub>y</sub> and aerosols. Measurement of the Essential Climate Variables (ECVs) pertaining to atmospheric composition, as defined by GCOS (Global Climate Observing System) in 2010 as
- necessary in order to understand the complex feedback mechanisms of the climate system.

Some recent studies have utilized measurements made on board of commercial aircrafts in order to better understand their impact on the dynamics of the carbon cycle. Niwa et al. 2012 examined the impact of passenger aircraft based on

- 35 measurements from CONTRAIL (Comprehensive Observation Network for Trace gases by Airliner) on the overall carbon budget constraint and the flux uncertainties. Patra et al. (2011) used measurements from the CARIBIC (Civil Aircraft for the Regular Investigation of the atmosphere Based on an Instrument Container) project as well as the CONTRAIL project to estimate regional CO<sub>2</sub> fluxes in the tropics. Both studies focused specifically on the estimation of the tropical terrestrial fluxes using mostly the free tropospheric part of the aircraft profiles. Gloor et al (2000) used aircraft vertical profiles in their
- 40 studies for observing network extension. A difference between fluxes estimated using near-surface observations and column average of the aircraft vertical profiles was discussed by Nakatsuka and Maksyutov (2009). However, so far, the suitability of

aircraft vertical profiles and their treatment when using them into inversions, given the transport modelling errors related to vertical mixing has not been addressed.


In this paper we employ synthetic data to investigate theoretical impacts of transport model uncertainties associated with boundary layer height on the fluxes retrieved by using passenger aircraft profiles in an inverse modelling set-up. The synthetic data are generated using a forward run of the TM3 transport model (Heimann and Körner, 2003) and have the temporal and spatial sampling of the measurements made during the MOZAIC project. We examine how closely the posterior flux obtained using the synthetic aircraft measurements as constraint captures the trends and variability in the flux that is used to generate the synthetic data. This allows us to estimate the impact of the inaccurate, simulated vertical mixing.

In the second part of this work, we assess the potential of CO<sub>2</sub> observations that will be onboard the IAGOS fleet for constraining the regional carbon budget and reducing posterior flux uncertainties. We further identify the regions that will benefit the most from these measurements. Only the time, location and uncertainty of the measurements are used for the simulations. Since flight routes of commercial aircraft undergo little changes with time, it is reasonable to estimate the constraint that will be brought about by IAGOS aircraft using the sampling from MOZAIC, its predecessor project.

#### 15

The paper is organized as follows: Section 2 describes the methods used that include estimation of the model representation error (Section 2.1), description of the inversion scheme (Section 2.2) and the experimental set-up (Section 2.3). Section 3 presents the results from the simulations and the conclusions are discussed in Section 4.

## 20

#### 2 Method

#### 2.1 Model description

#### 2.1.1 Inversion principle

The Jena inversion system (Roedenbeck et al. 2005) is a Bayesian inversion framework that is used to estimate trace gas fluxes at the surface of the earth from measured atmospheric concentrations and knowledge of atmospheric transport. It employs the global atmospheric tracer model TM3 to simulate atmospheric transport (Heimann and Körner, 2003). In this study, our model simulations are carried out at a 4°×5° spatial resolution using the ERA-Interim (European Centre for Medium Range Weather Forecasts (ECMWF) Reanalysis-Interim) meteorology.

In the following paragraphs, we provide a brief description of the inversion system described in more detail in Roedenbeck et al. (2005). Observed atmospheric mixing ratios  $C_{obs}$ , are compared to modelled atmospheric mixing ratios,  $C_{mod}$ , based on a prior estimate of the surface fluxes. The modelled atmospheric mixing ratio at a specific location,  $C_{mod}$  is obtained by the multiplication of the linear atmospheric transport operator A computed by the transport model with the flux field **f** and the addition of the initial atmospheric mixing ratio of the transport model at the beginning of the simulation period,  $C_{ini}$ 

$$\mathbf{C}_{\mathrm{mod}} = \mathbf{A}\mathbf{f} + \mathbf{C}_{\mathrm{ini}} \tag{1}$$

The concentration mismatch between observed and modelled values is defined as

$$\mathbf{m} = \mathbf{C}_{\rm obs} - \mathbf{C}_{\rm mod} \tag{3}$$

The aim of the inversion system is to optimize the conditional (a posteriori) probability of the model parameters **p** with respect to the **m**, according to Bayes' Theorem. This corresponds to minimising the cost function **J** defined as:

$$\mathbf{J} = -\ln\left(\operatorname{Prob}\left(\mathbf{p} \mid \mathbf{m}\right)\right) \tag{5}$$

$$=\frac{1}{2}\mathbf{m}^{\mathrm{T}}\mathbf{Q}_{\mathrm{c}}^{-1}\mathbf{m}+\frac{\mu}{2}\mathbf{p}^{\mathrm{T}}\mathbf{p}+\mathsf{C}$$
(6)

5 The difference between the modelled C<sub>mod</sub> and observed C<sub>obs</sub>, **m** is used to calculate the observation-based term of a cost function which forms the first term of Eq. (6); taking into account the measurement and model representation errors. <sup>μ</sup>/<sub>2</sub> **f**<sup>T</sup><sub>post</sub> **f**<sub>post</sub> describes the a-priori flux constraints. The additive constant C subsumes all parameter independent terms, such as those arising from Prob (**m**) and from the normalization of the distribution. This cost function is minimized iteratively using the adjoint of the atmospheric transport model, as the number of observations and variables to constrain is very large, therefore prohibiting the calculation of an analytical solution. **Q**<sub>c</sub> is defined as the error covariance matrix of the atmospheric mixing ratio mismatch. Its diagonal elements represent the combined measurement and modelling errors for each observation i.e. σ<sub>*i*,tot</sub> =  $\sqrt{\sigma_{mod}^2 + \sigma_{meas}^2}$ . In order to scale the impact of the a-priori constraint on the Bayesian inversion the factor **μ** is used. It determines the ratio between the a-priori information and data constraints. For **μ** equal to 0 no prior information is used for minimizing the cost function. For high values of **μ** the a-priori flux distribution has a high impact on the minimization of the cost function.

#### 2.1.2 Data density de-weighting

- 20 The existing observation network consists of a number of ground-based stations that measure at different temporal frequencies. While stations based on flask observations have measurements made once per day or once per week, there also exist a growing number of continuously measuring stations with data provided typically half hourly or hourly. For the aircraft profiles, the profile measurements are made over a period of approximately 30-40 minutes during the ascent or descent of the aircraft. Therefore many of the measurements made by surface stations in a single day or in a single aircraft profile cannot be treated as independent of each other. This means that the errors of such measurements are likely to be correlated with each other over certain temporal scales. To account for this fact in the simulations, the error of correlated measurements is enhanced (or "inflated"), so that their contribution to the cost function is reduced. In this way the impact of continuous observations from a single station has a comparable impact on the cost function as less frequent flask observations from another station.

In the Jena inversion scheme, these error correlations between measurements are accounted for using a data density 'deweighting' scheme. It assigns a weight to the error associated with every measurement computed based on certain predefined criteria. For surface network sites, to avoid a higher impact of the more frequent continuous observations compared to the less frequent flask observations, the data density weighting considers, for every observation, the number of observations  $N_{surf}$  within the same week. The total uncertainty for that observation increases by a factor of  $\sqrt{N_{surf}}$ . These  $N_{surf}$  measurements have their errors correlated and this error inflation by a factor of  $\sqrt{N_{surf}}$  helps lessen the impact of measurements that are not independent of each other and hence their contribution to the cost function.

The aircraft is a moving platform, which means that the aircraft profiles span a considerable horizontal and vertical distance while making measurements. Therefore, in contrast to a fixed station, the  $CO_2$  concentration along the profile can be expected to de-correlate due to distance, even if taken within a short period of time. We need to incorporate this fact in the de-weighting scheme. Thus, for the aircraft profiles,  $N_{aircraft}$  is defined to be the number of measurements that lie in a 4-D (3D space and time) window instead of just those lying within a 1-week interval as used for the surface stations. Measurements that lie within this 4-D window are taken to have their errors correlated with each other, but taken independent of those that lie outside of it. The 4-D space is defined using the following criteria:

- 1. Temporal de-correlation length is taken to be 1 week, to be consistent with the treatment of the station data.
- 2. Horizontal spatial de-correlation distance is set at +/-500 km for measurements within the first 700 mbar from the surface and +/-1000 km for the ones above the 700 mbar height.

We use these values of spatial correlation lengths since they are comparable to the grid size that we use for our simulations and sub-grid scale processes cannot be resolved by the transport model. The 700 mbar pressure level represents approximately the maximum of a typical boundary layer height and separates the boundary layer part of the atmospheric column (which is more closely coupled to surface fluxes by fast vertical mixing and hence has a shortened correlation length) from the free troposphere part of the column.

10

15

5

## 2.2 Estimation of model data mismatch error

Model representation error or model-data mismatch can be defined as the mismatch between point observations assimilated in the model and the model simulated spatial averages (Engelen et al. 2002). This error needs to be pre-specified in inversion framework. In our model, we use a representation error that varies with altitude. This is because the mismatch is likely to be higher for measurements that lie closer to the surface while the models perform better for higher altitudes that are not affected as directly by the fluxes. The functional dependency of the mismatch with altitude is computed using data from the CONTRAIL project (Machida et al. 2008).

We compute the dependency of the mismatch on altitude using data from the CONTRAIL project (Machida et al. 2008). For this, we compare observations from CONTRAIL against TM3 "reanalysed CO<sub>2</sub> fields" (i.e., atmospheric CO<sub>2</sub> fields simulated by the tracer transport model from surface fluxes previously optimized against CO2 data, such that these fields closely match the data and interpolate in between them). The difference gives the model-data mismatch (*mdm*) at every level for each airport where CONTRAIL aircraft fly. The vertical resolution of CONTRAIL is 0.25 km, however the statistics have been aggregated onto a coarser 1-km resolution for this analysis. In order to obtain a typical *mdm* at every level of a profile we use the median of the standard deviation of the *mdm* at each level across all airports that have at least 20 data points. Figure 1 shows a box plot that is thus obtained. We then fit an exponential curve to the median values at each level:

$$mdm = ae^{bz} + c \tag{7}$$

where we obtain a= 2.85 ppm, b= -0.4, and c= 3.18 ppm.

#### 2.3 Experimental Setup

- Synthetic data at the times and locations of the MOZAIC profiles and the ground network sites are generated to both investigate the impact of boundary layer height errors and assess the impact the addition of aircraft observations has on flux retrievals. For the forward run, we use fluxes from the BIOME-BGC biosphere model (Thornton et al., 2005) in order to get realistic mixing ratios at the locations of aircraft profiles and the surface stations. These fluxes form our "true flux". The MOZAIC aircraft profiles consist of measurements provided at approximately every 150 m altitude starting at 75 m and
- going typically up to an altitude of 9-10 km. We choose not to use the cruise level data for this study because of the fact that most of these measurements are made around the tropopause region, and the model skill in accurately representing the


transport at that altitude and linking those measurements via vertical transport to fluxes at the surface is limited (Deng et al. 2015)

Since the profiles generated by the forward run of the transport model use the ERA-interim meteorology, the boundary layer height represented by these profiles is that of ERA-interim. We call this the "true" boundary layer height,  $BLH_{true}$  In order to

- 5 simulate the vertical-mixing-related imperfections in the transport models, we need to generate new profiles with a "wrong" boundary layer height. We do this by modifying these profiles in such a way that they represent a new boundary layer height that is different from  $BLH_{true}$ .  $BLH_{model}$  denotes this "wrong" boundary layer height. In order to achieve this we use the approach as implemented by Kretschmer et al., 2012. This approach assumes that errors in the simulated boundary layer height are caused by incorrect vertical distribution of  $CO_2$  in a given atmospheric column, such that the total column
- 10 concentration remains unchanged. We redistribute the  $CO_2$  between the free troposphere and boundary layer part of the atmospheric column in such a way that the BLH for the profile changes to  $BLH_{model}$ . In this study, we use the  $BLH_{model}$  obtained from the NCEP (National Centres for Environmental Prediction) meteorology. In order to compute the boundary layer height the Bulk Richardson Number method was used,

The effect of vertical mixing errors in transport models on flux retrieval is analysed with three groups of experiments:

S: Simulation with only the surface-based observation network.

A: Simulation using only the IAGOS aircraft profiles.

C: Simulation with the combined network: surface-based observation network augmented with the measurements from IAGOS.

#### 20


For each of these simulations, we further carry out two types of inversions:

- a. Original profiles (Control case)
- b. Reshuffled profiles.

Experiments S (a), A (a) and C (a) represent scenarios where the boundary layer height is well known. Experiments S (b), A
(b) and C (b) simulate the realistic case where the vertical mixing in the transport model is imperfect. The monthly posterior fluxes are analysed for one year (2000). The surface network consists of 49 sites (Fig. 2(a)) and the IAGOS observation network consists of measurements from five IAGOS aircraft (Fig. 2(b)). The prior flux used for the inverse simulations is different and independent from the true flux used to generate the pseudo data and is obtained from the Lund-Potsdam-Jena (LPJ) dynamic global vegetation model (Sitch et al., 2003)



In the second part of the study, we estimate the reduction in posterior flux uncertainty brought about by the use of IAGOS vertical profiles as a constraint on the carbon budget. We carry out simulations where the surface-based observation network is augmented by one or more IAGOS aircraft. These simulations do not require the synthetic data that as used in the first part of this study since the inversion system solves for the resultant posterior flux uncertainties based upon only the measurement time, location and the uncertainties of the prior fluxes and the measurements (model-data mismatch). The uncertainty reduction is computed for the monthly mean posterior fluxes aggregated over the TransCom3 land regions (Gurney et al., 2000). It is expressed as the following:

Uncertainty Reduction ( in percent) = 
$$\left(1 - \frac{\text{posterior uncertainty}}{\text{prior uncertainty}}\right) \times 100\%$$
 (8)


It is defined as the extent to which the error in the flux field is modified by the inversion. It is dependent on both the prior uncertainty as well as the observation coverage and is a measure of the accuracy of the posterior fluxes estimated by the inversion.

Figure 3 shows the prior uncertainty used by the Jena inversion scheme for the different TransCom3 regions. We focus on the years 1996-2004 because of sufficient data availability from MOZAIC during this period. This period also has some data gaps representing times when one or more aircraft are not flying. This helps give a more realistic quantification of the uncertainty reduction brought about by the use of these data.

#### **3** Results and Discussion

#### 3.1 Impact of BLH transport model errors on flux retrieval

We analyse monthly posterior fluxes for the TransCom3 land regions and compare them to our "true" flux, which is the flux that is used to generate our pseudo data. We concatenate the time series of the posterior flux for all regions to form a single time series in order to obtain a single diagnostic metric for the whole globe. The statistics for comparison between the different simulations are represented on a Taylor diagram as shown in Fig. 4.

We see that the transport model errors related to vertical mixing, as simulated using the reshuffling method, affect the flux

- retrieved from measurements made at surface stations differently than those retrieved using aircraft profiles. We observe that there is a large impact of the simulated vertical mixing errors on the flux retrieved using the surface measurements with
- that there is a large impact of the simulated vertical mixing errors on the flux retrieved using the surface measurements with and without the boundary layer height uncertainties incorporated in the experiments as shown by points Sb and Sa respectively. The posterior flux standard deviation, root-mean-square difference and correlation coefficient values with respect to the true flux change from 1.90 PgC year<sup>-1</sup>, 0.65 PgC year<sup>-1</sup> and 0.95 respectively for the simulation Sa to 2.39 PgC year<sup>-1</sup>, 1.76 PgC year<sup>-1</sup> and 0.69 for simulation Sb. On the other hand, the erroneous vertical mixing has nearly no impact on
- the flux retrieval using aircraft profile measurements. The standard deviation, root-mean-square difference and correlation coefficient values of 1.81 PgC year<sup>-1</sup>, 0.70 PgC year<sup>-1</sup>, 0.94 change only marginally to 1.82 PgC year<sup>-1</sup>, 0.72 PgC year<sup>-1</sup> and 0.94 shown by the overlap of the points Aa and Ab. This difference in the response of the flux retrieved using observations from the two different measurement platforms to vertical mixing error can be explained as follows: The aircraft profiles, by virtue of their vertical extent, constrain the inversion using observations at nearly all tropospheric layers over which the total
- column CO<sub>2</sub> abundance remains constant since CO<sub>2</sub> is well mixed in the troposphere. The impact of vertical transport near the surface is solely to redistribute the tracer mass in the atmospheric column between the different layers of the atmosphere, keeping the total column abundance unchanged. In other words, due to this redistribution, the loss of tracer mass in the boundary layer is compensated by the gain in mass in the free troposphere and vice versa. Therefore, any change in the vertical distribution of the tracer at these levels is not likely to impact the total tracer mass in the profile that constrains the
- inversion and hence the resultant posterior flux retrieved using these measurements. The surface station measurements, on the other hand, are made at a single altitude, generally within the boundary layer and hence, any change in the boundary layer height due vertical mixing, is likely to cause an impact on the modelled mixing ratio at the measurement altitude which is used to constrain the inversion and hence in the flux retrieved by the inversion. The posterior flux shows less sensitivity to boundary layer height errors in the transport model when aircraft profiles are used as constraint while surface measurements
- are more likely to be affected by these errors, which translates into errors in the retrieved flux. Points Ca and Cb in Fig. 4 show the impact of the boundary layer error on the flux retrieved using the combined observation network that uses measurements from both the surface network and the aircraft profiles. By using the combined observation network, a similar sensitivity of the posterior flux to boundary layer uncertainty is observed as by the surface based network alone (Points Sa and Sb). This similarity in sensitivity of posterior flux between simulation types C and S shows that the
- effect of the surface network dominates the flux retrieval from the observations using the combined network and indicates that the surface network stations largely contribute to the sensitivity of the retrieved flux to the uncertainty of the boundary layer height. It can also be seen that the addition of aircraft measurements leads to an improved estimate of the surface flux. This is shown by points Ca and Cb being closer to the true flux than points Sa and Sb respectively. It implies that the

addition of the aircraft measurements to the surface based network improves the constraint on the carbon budget as compared to the surface network alone.

#### 1.2 Constraint on carbon budget due to IAGOS aircraft profiles

In this section, we evaluate the utility of aircraft measurements of  $CO_2$  from IAGOS for constraining the regional carbon budget. For this the reduction in the uncertainty of the posterior fluxes in relation to the prior fluxes is assessed. It should be noted that while the uncertainty reduction alone may not be robust, similarly computed uncertainty reductions can be robustly compared.

- Figure 5(a) shows the flux uncertainty reduction of the monthly mean flux over the TransCom3 regions when only the surface based observational network is used in the inversion. The largest constraint due to the surface network alone is observed in Europe and North America. The European and Temperate North American regions have a dense and extensive network of surface observations and hence the reduction in flux uncertainty is as high as about 85 %. In addition, remote observations are also responsible for bringing about a constraint on the fluxes in the neighbouring regions due to the effect of
- wind transport (horizontal advection). For instance, the value of the uncertainty reduction over North American boreal regions (75 %) is high inspite of insufficient surface stations in that region. This can be attributed to the impact of the westerly winds flowing over Temperate North America. West winds mean that observations in these regions are sensitive to boreal fluxes. Using the same argument, dense observations over Europe can help constrain surface fluxes from the Eurasian boreal region due to the effect of transport (advection) by the westerlies.
- Figure 5(b) shows the uncertainty reduction only due to the pseudo profiles from five simulated IAGOS aircraft. Europe, temperate North American regions show an uncertainty reduction of about 70 %. These regions have a where most of the aircraft profiles are measured due to large air traffic between the two continents by the airlines participating in MOZAIC/IAGOS. These measurements are also able to constrain boreal North America (70 %) and boreal Eurasia (55 %), regions with little or no MOZAIC/IAGOS measurements. The African continent shows a high reduction in flux uncertainty
- (75 %). Regions of South America and Tropical Asia exhibit a low constraint ranging between 20 % and 35 %, due to fewer aircraft profiles measured in these regions in addition to the impact of advection by the easterly winds. Figure 5(c) shows the uncertainty reduction map for the case when pseudo profiles IAGOS aircraft are added to the surface based network. The combined observation network almost completely constrains the regions of Europe and Temperate North America, the uncertainty reduction value being close to 90 %. Tropical Asia is the least constrained by the combined
- network since it is not adequately covered by either of the networks- surface or the passenger aircraft. The net impact of adding the profiles from IAGOS to the existing network is shown in Fig. 5(d), which is the difference between the uncertainty reduction values for the TransCom3 land regions with and without the aircraft profiles. Tropical and Eurasian temperate regions show the greatest change in the uncertainty reduction of the posterior fluxes on addition of pseudo observations from IAGOS (about 7 to 10 %). These are regions that are poorly constrained by the surface based network. So,
- addition of aircraft measurements results in the largest improvement in posterior flux uncertainty in these regions. On the other hand, for regions already well constrained by the surface network, for example North America and Europe, the simulated constraint due to the IAGOS CO<sub>2</sub> measurements is very small (less than 1 %).

We further investigated the constraint due to the aircraft measurements on aggregated spatial scales by examining the change in uncertainty reduction on the addition of pseudo measurements from IAGOS for the Northern hemisphere (30° N to 90°

40 N), Tropics (-30° S to 30° S) and Southern hemisphere (-90° S to -30° S). The zero measurements point on the x-axis of Fig. 6(a) and 6(b) indicates the case where only the existing observation network sites have been used into the inversion but no IAGOS profiles have been used. The change in the uncertainty reduction for the northern hemisphere posterior uncertainty increases from 0.5 % when measurements from one simulated IAGOS aircraft are used, to 2 % from measurements from five

aircraft. The Tropics, on the other hand, show a comparable trend and increase in the change of flux uncertainty with up to 10 times fewer measurements than in the Northern hemisphere. This difference in uncertainty reduction change in the two regions is likely to be due to the fact that unlike the Northern hemisphere, the tropics are not well constrained by the existing network. Hence, the addition of IAGOS profiles leads to considerable constraint on the surface fluxes. The southern hemisphere (not shown), which is largely ocean, does not gain much from these measurements since they are very few in number and are not sufficient to constrain the region. Hence almost no change is seen in the uncertainty reduction due to aircraft measurements. Thus, we can conclude that the overall impact of IAGOS measurements based upon this sampling is highest for the tropical region. This indicates that the greatest incremental increase in knowledge of fluxes would be gained by instrumenting aircraft flying preferentially tropical routes. It is however noteworthy, that the saturation of posterior

10

5

uncertainty values as the number of measurements approaches the maximum value, does not imply that there would be no further benefit of adding measurements from more than from five aircraft. The figure is indicative of the information gained solely on aggregated spatial scales and it is very likely that on smaller scales there is added benefit of having more measurements.

15

## 4 Summary

Transport models that drive the inversion schemes often have a poor representation of the near surface vertical mixing 20 causing large errors in the retrieved fluxes. In this study, we investigate the impact of such transport model uncertainties on the fluxes simulated using aircraft profiles as constraint in an inverse modelling set up. We focus only on errors in nearsurface vertical mixing. Those due to imperfect representation of other processes like advection and deep convection have not been accounted for. Our simulations show that the flux retrieved using aircraft profiles when the boundary layer height is well known has the same statistical metrics as the flux retrieved when the boundary layer height is erroneous. This shows 25 that posterior fluxes retrieved using aircraft profiles show no sensitivity to the boundary layer height errors as simulated in our experiments. We compare this behaviour of the retrieved flux to that obtained using the surface measurements as constraint. These measurements are usually in the boundary layer part of the atmosphere and therefore we find a much higher mismatch between the flux retrieved using correct versus erroneous boundary layer height in terms of the standard deviation, root-mean-square difference and correlation parameters. In other words, this mismatch shows that the transport 30 model uncertainties related to boundary layer height are very likely to be translated to the posterior flux when surface measurements are used as constraint in the inversion while these errors are not propagated to the retrieved flux when the aircraft profiles are used. This difference in the response of the flux retrieved using the two observation networks is likely to be due to the fact that vertical transport, whose effect we simulate by the redistribution of the tracer mass in the model profile at the location of the airports and surface stations, only redistributes the tracer mass between the boundary layer 35 height and the free tropospheric part keeping the total tracer mass constant. The loss (or gain) of the tracer mass in the profile in the boundary layer part of the profile is compensated by the gain (or loss) in the free tropospheric part of the profile. Since aircraft profile measurements extend all the way from the surface to the free tropospheric part of the atmosphere, the net impact of the complete reshuffled profile remains comparable to that of the original. This effect of redistribution, on the

40 error in the estimation of the boundary layer height will impact the modelled mixing ratio that constrains the inversion. These results demonstrate the benefit of aircraft measurements over those made by ground-based stations for flux estimation using transport models that cannot resolve the boundary layer perfectly. Although we only account for errors in fluxes due to vertical mixing in our simulations, we can say that flux estimation using aircraft profiles is expected to be more robust when

other hand, is not observed for the surface station measurements since they are made within the boundary layer and hence,

aircraft profiles are used as constraint since the contribution of the boundary layer height uncertainty to the overall transport model error is likely to decline. While improved transport models are an imperative for achieving more accurate estimates of surface fluxes, the potential benefit of aircraft profiles over ground-based measurements, as shown by our simulations, provides a simple and flexible approach of dealing with and eliminating the impact of boundary layer height uncertainties due to vertical mixing and diminishing the overall impact of transport model errors on retrieved fluxes. In addition to this,

- aircraft profiles would also provide valuable information to drive model development. Furthermore, on estimating the impact that the  $CO_2$  measurements made onboard the IAGOS fleet are likely to have on the regional carbon budget once they are available, we find that the IAGOS flights will likely provide a strong constraint on regional flux totals. The net  $CO_2$  flux uncertainty reduction using the IAGOS measurements is likely to be highest in the
- Tropics and the Eurasian temperate regions. These are regions that are not well covered by the existing surface based observation network and hence the addition of aircraft measurements brings about the largest constraint. The change in the uncertainty reduction in these regions is between 7 to 10 percent. In contrast, the European and North American continents, which have good data coverage by the surface, based network show little or no change in flux uncertainty due to added measurements from IAGOS.
- We must bear in mind that since the MOZAIC/IAGOS aircraft profiles are measured near the airports, which form areas of high anthropogenic emissions, it is likely that these observations are not truly representative of large areas. This fact has been taken into account, in this study, in a conservative way by estimating the model data mismatch uncertainty using the difference between CO<sub>2</sub> profiles from the CONTRAIL project and reanalysed TM3 fields (Sect. 2.2). However, better approaches for addressing this question of representativeness of aircraft profiles exist, for example, those described by
- Boschetti et al. 2015. A relatively high model-observation mismatch of 5 ppm at 1 km (as shown in Fig. 1) for CONTRAIL data could partly be a result of applying low-resolution (with respect to plume size of anthropogenic CO<sub>2</sub> transported from large cities near the airports) model and meteorology and thus should be considered as an upper bound on the model data mismatch.
- In summary, our results demonstrate the benefit and application of aircraft profile measurements in an inverse modelling framework. In the near future, increased number aircraft profiles of greenhouse gases are expected to be available. Hence, exploiting the potential advantage of this new data stream for inverse modelling studies can go a long way to developing a better understanding of carbon cycle dynamics in hitherto under-sampled regions of the world.

5

#### Acknowledgements:

The research leading to these results has received funding from the European Community's Seventh Framework Programme ([FP7/2007-2013]) under grant agreement n° 312311 for the IGAS project (IAGOS for the GMES Atmospheric Service)

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

## 15

Denning, A. S., Zhang, N., Yi, C. X., Branson, M., Davis, K., Kleist, J., and Bakwin, P.: Evaluation of modeled atmospheric boundary layer depth at the WLEF tower, Agr. Forest Meteorol.,148, 206–215, 2008.

Ehret, G. and Kiemle, C.: Requirements definition for future DIAL instruments, Study report ESA-CR(P)-4513, ESA, Noordwijk, The Netherlands, 2005.

Engelen, R. J., Denning, a S. and Gurney, K. R.: On error estimation in atmospheric CO2 inversions, J. Geophys. Res. Atmos., 107, ACL 10–1–ACL 10–13, doi:10.1029/2002JD002195, 2002.

Galli, A., Guerlet, S., Butz, A., Aben, I., Suto, H., Kuze, A., Deutscher, N. M., Notholt, J., Wunch, D., Wennberg, P. O., Griffith, D. W. T., Hasekamp, O. & Landgraf, J.: The impact of spectral resolution on satellite retrieval accuracy of CO<sub>2</sub> and CH<sub>4</sub> Atmospheric Measurement Techniques, 7(4), 1105–1119. doi:10.5194/amt-7-1105-2014, 2014.

Gerbig, C., Lin, J. C., Wofsy, S. C., Daube, B. C., Andrews, a. E., Stephens, B. B., Bakwin, P. S. and Grainger, C. a.: Toward constraining regional-scale fluxes of CO<sub>2</sub> with atmospheric observations over a continent: 1. Observed spatial variability from airborne platforms, J. Geophys. Res. Atmos., 108(D24), n/a–n/a, doi:10.1029/2002JD003018, 2003.

Gerbig, C., Körner, S., and Lin, J. C.: Vertical mixing in atmospheric tracer transport models: error characterization and propagation, Atmos. Chem. Phys., 8, 591-602, doi:10.5194/acp-8-591-2008, 2008.

Gloor, M., S.-M. Fan, S. Pacala, and J. Sarmiento, Optimal sampling of the atmosphere for purpose of inverse modeling: A model study, Global Biogeochem. Cycles, 14(1), 407–428, doi:10.1029/1999GB900052, 2000.

Gurney, K. R., Law, R. M., Denning, a S., Rayner, P. J., Baker, D., Bousquet, P., Bruhwiler, L., Chen, Y.-H., Ciais, P., Fan, S., Fung, I. Y., Gloor, M., Heimann, M., Higuchi, K., John, J., Maki, T., Maksyutov, S., Masarie, K., Peylin, P., Prather, M., Pak, B. C., Randerson, J., Sarmiento, J., Taguchi, S., Takahashi, T. and Yuen, C.-W.: Towards robust regional estimates of CO2 sources and sinks using atmospheric transport models., Nature, 415(February), 626–630, doi:10.1038/415626a, 2002.

Gurney, K., Law, R., Rayner, P., and A.S. Denning, "TransCom 3 Experimental Protocol," Department of Atmospheric Science, Colorado State University, USA, Paper No. 707, 2000.

Haszpra, L.: On the representativeness of carbon dioxide measurements, J. Geophys. Res. Atmos., 104(D21), 26953–26960, doi:10.1029/1999JD900311, 1999.

Heimann, H., Körner, S.: The global atmospheric tracer model TM3. Technical Reports - Max-Planck-Institut für Biogeochemie 5, pp. 13, 2003


Houweling, S., Aben, I., Breon, F.-M., Chevallier, F., Deutscher, N., Engelen, R., Gerbig, C., Griffith, D., Hungershoefer, K., Macatangay, R., Marshall, J., Notholt, J., Peters, W., and Serrar, S.: The importance of transport model uncertainties for the estimation of CO<sub>2</sub> sources and sinks using satellite measurements, Atmos. Chem. Phys., 10, 9981–9992, doi:10.5194/acp-10-9981- 2010, 2010.

Law, R. M., Rayner, P. J., Denning, A. S., Erickson, D., Fung, I. Y., Heimann, M., Piper, S. C., Ramonet, M., Taguchi, S., Taylor, J. A., Trudinger, C. M. and Watterson, I. G.: Variations in modeled atmospheric transport of carbon dioxide and the consequences for CO2 inversions, Global Biogeochem. Cycles, 10(4), 783–796, doi:10.1029/96GB01892, 1996.

Law, R. M., Matear, R. J., and Francey, R. J.: Saturation of the Southern Ocean CO2 sink due to recent climate change, Science, 319, 570a–570a, 2008.

Lauvaux, T., Pannekoucke, O., Sarrat, C., Chevallier, F., Ciais, P., Noilhan, J. and Rayner, P. J.: Structure of the transport uncertainty in mesoscale inversions of CO2 sources and sinks using ensemble model simulations, Biogeosciences, 6(6), 1089–1102, doi:10.5194/bg-6-1089-2009, 2009.

Machida, T., Matsueda, H., Sawa, Y., Nakagawa, Y., Hirotani, K., Kondo, N., Goto, K., Nakazawa, T., Ishikawa, K. and
 Ogawa, T.: Worldwide Measurements of Atmospheric CO2 and Other Trace Gas Species Using Commercial Airlines, J. Atmos. Ocean. Technol., 25, 1744–1754, doi:10.1175/2008JTECHA1082.1, 2008.

Marquis, M. and Tans, P.: CLIMATE CHANGE: Carbon Crucible, Science (80-. )., 320(5875), 460-461, doi:10.1126/science.1156451, 2008.

Marenco, A., Thouret, V., Nédélec, P., Smit, H., Helten, M., Kley, D., Karcher, F., Simon, P., Law, K., Pyle, J., Poschmann,
 G., Von Wrede, R., Hume, C. and Cook, T.: Measurement of ozone and water vapor by Airbus in-service aircraft: The MOZAIC airborne program, an overview, J. Geophys. Res., 103(D19), 25631, doi:10.1029/98JD00977, 1998.

Nakatsuka, Y. and Maksyutov, S.: Optimization of the seasonal cycles of simulated CO<sub>2</sub> flux by fitting simulated atmospheric CO<sub>2</sub> to observed vertical profiles, Biogeosciences, 6, 2733-2741, doi:10.5194/bg-6-2733-2009, 2009.

Niwa, Y., Machida, T., Sawa, Y., Matsueda, H., Schuck, T. J., Brenninkmeijer, C. A. M., Imasu, R. and Satoh, M.: Imposing strong constraints on tropical terrestrial CO2 fluxes using passenger aircraft based measurements, J. Geophys. Res. Atmos., 117(11), doi:10.1029/2012JD017474, 2012.

Patra, P. K., Niwa, Y., Schuck, T. J., Brenninkmeijer, C. A. M., MacHida, T., Matsueda, H. and Sawa, Y.: Carbon balance of South Asia constrained by passenger aircraft CO 2 measurements, Atmos. Chem. Phys., 11(9), 4163–4175, doi:10.5194/acp-

Rödenbeck, C., Houweling, S., Gloor, M. and Heimann, M.: CO2 flux history 1982–2001 inferred from atmospheric data using a global inversion of atmospheric transport, Atmos. Chem. Phys., 3(6), 1919–1964, doi:10.5194/acp-3-1919-2003, 2003.




Rödenbeck, C. : Estimating CO2 sources and sinks from atmospheric mixing ratio measurements using a global inversion of atmospheric transport. Technical Report 6, Max Planck Institute for Biogeochemistry, Jena, 2005.

Sitch, S., Smith, B., Prentice, I. C., Arneth, A., Bondeau, A., Cramer, W., Kaplan, J. O., Levis, S., Lucht, W., Sykes, M. T., Thonicke, K. and Venevsky, S.: Evaluation of ecosystem dynamics, plant geography and terrestrial carbon cycling in the LPJ dynamic global vegetation model, Glob. Chang. Biol., 9(2), 161–185, doi:10.1046/j.1365-2486.2003.00569.x, 2003.

Stephens, B. B., Gurney, K. R., Tans, P. P., Sweeney, C., Peters, W., Bruhwiler, L., Ciais, P., Ramonet, M., Bousquet, P., Nakazawa, T., Aoki, S., Machida, T., Inoue, G., Vinnichenko, N., Lloyd, J., Jordan, A., Heimann, M., Shibistova, O., Langenfelds, R. L., Steele, L. P., Francey, R. J. and Denning, A. S.: Weak northern and strong tropical land carbon uptake from vertical profiles of atmospheric CO2., Science, 316(5832), 1732–5, doi:10.1126/science.1137004, 2007.

Thornton, P. E., S. W. Running, and E. R. Hunt. : Biome-BGC: Terrestrial Ecosystem Process Model, Version 4.1.1. Data model. Available on-line [http://www.daac.ornl.gov] from Oak Ridge National Laboratory Distributed Active Archive Center, Oak Ridge, Tennessee, U.S.A., doi:10.3334/ORNLDAAC/805, 2005

Yi, C., Davis, K. J., Bakwin, P. S., Denning, A. S., Zhang, N., Desai, A., Lin, J. C. and Gerbig, C.: Observed covariance between ecosystem carbon exchange and atmospheric boundary layer dynamics at a site in northern Wisconsin, J. Geophys. Res. D Atmos., 109(8), doi:10.1029/2003JD004164, 2004.

Figure 1: Box plot showing the model data mismatch between the TM3 analysed  $CO_2$  fields and the vertical profiles from the CONTRAIL project plotted against height. The red line shows the exponential curve fitted to the median of the standard deviation of the model data mismatch.