# Peer review of "The constraint of CO2 measurements made onboard passenger aircraft on surfaceatmosphere fluxes: the impact of transport model errors in vertical mixing"

_Atmospheric Chemistry and Physics, 2016_

## Referee Comment (RC1) · Anonymous Referee #2 · 31 Aug 2016

The authors have written a wonderful paper on the utility of CO2 observations from commercial aircraft and the impacts of uncertainties in vertical transport. I have listed a few suggestions below to consider during the revision process.

Pg 1, line 14: I would include a noun after a word like "this" or "these" so it is clear to the reader what you are referring to in the sentence. For example, "This [fill in here] highlights ...."

Pg 1, line 14: consider changing "the benefit" to "a benefit"

Pg 1, line 20: Suggested rewording: "with a reduction of posterior flux uncertainty of

about 7 to 10%."

Pg 1, line 22: add a comma before "and"

Pg 1, line 24: What does "these" refer to here? I would include a noun after "these".

Pg 1, line 28-29: Consider adding references to this sentence.

Pg 1, line 35: Consider adding references to this sentence.

Pg 2, lines 2-4: The observation network could be spatially uneven and could be temporally sparse or uneven (e.g., as is the case with some aircraft campaigns). This point could be worth mentioning here, depending on whether you think it flows with the text.

Pg 2, line 11-12: This point is true of short towers but not necessarily of tall towers (e.g., the NOAA tall tower network in the US).

Pg 2, line 14: Instead of "constraints", I would use a word like "limitations" ("... they have their limitations, too, which restricts their use for accurate flux ...."). The word "constraints" makes it sound like the satellite is constraining something (e.g., fluxes), not that the satellite has limitations.

Pg 2, line 41: Suggested rewording: "... investigate theoretical impacts of transport model ...."

Pg 2 lines 10-17: Some of the information here seems redundant with information in the previous two paragraphs. You may want to eliminate these lines or condense with the previous two paragraphs.

Pg 2, line 2: I prefer active voice (E.g., "section 3 presents results") over passive voice ("In section 2, the results are presented"). This wording could be a matter of personal preference.

Pg 2, line 29: A 4x5 degree resolution seems really coarse for CO2 simulations. At this juncture, I imagine it would be difficult to find a higher resolution model and re-run all of

the CO2 model simulations. However, somewhere in the paper (e.g., in a supplement), it might be useful to explain why you used this resolution and how the resolution affects your interpretation of the results.

Pg 2, line 31: I would include a noun after "following."

Pg 4, Eq. 2: How does f_post relate to the variables in this equation? Is "f_post" the same as "f"?

Pg 4, Eq. 4b: Many inverse modeling studies include an a priori covariance matrix that defines uncertainties in the prior and spatial/temporal covariances in these covariances. Instead, you have used a weighting factor (mu). It could be helpful to explain why you have chosen the latter approach over the former.

Pg 4, line 12: The symbol mu usually refers to the mean in statistics. A different variable name here could prevent confusion. Section 2.1.1: This section contains a lot of information about the fundamentals of Bayesian inverse modeling, and it appears that a lot of this information has already been published elsewhere. You could either condense this section or move the material to a supplement.

Pg 4, line 38: Are there many stations that have weekly flasks? I know of a number of stations with daily flasks. You could change this sentence to "... made once per day or once per week ...."

Pg 5, lines 1-5: Why use this approach instead of including covariances or off-diagonal elements in Q?

Pg 5, lines 5-24: You could condense this information or move it to a supplement if it has been published elsewhere.

Pg 5, lines 37-38: This definition of model-data mismatch is confusing. For example, what simulated spatial averages are you referring to here? Does this definition include or exclude temporal averaging? Also, I think this definition of model-data mismatch is different than the definition used in other inverse modeling papers (e.g., Michalak et al.

2004, doi:10.1029/2003JD004422).

Pg 8, line 7: I would include a comma after "redistribution."

Pg 8, line 19: What does "this" refer to? I would add a noun after "this."

Pg 8, line 28: The word "constraint" is potentially confusing here. You could use a different phrase like "evaluate the utility of the aircraft measurements ...."

Pg 8, line 39: The phrase "cause a constraint" may not be the optimal wording here. Instead, you could use a phrase like the following: "West winds mean that observations in these regions are sensitive to boreal fluxes."

Pg 9, line 8: I would re-phrase "least constraint in the fluxes." Instead, you could try "least constrained by the combined network ...".

Pg 9, line 12: Throughout the article, the word "constraint" is used in a variety of different contexts with different meanings. I would use this word in a single context and choose different words in different contexts. In this line, I think the phrase "uncertainty reduction" might be more appropriate.

Pg 9, line 24: I would remove the word "however."

Pg 9, line 25-26: I think this sentence is a run-on. I would put a period after "network" and start a new sentence with the word "hence." Also, I would add a noun after the word "this".

Pg 10, lines 7 and 10: What do the words "this" refer to in each sentence? I would add a noun after the word "this" in each case.

Pg 10, line 22: I would change "reduce" to "decline".

Pg 10, line 28: This sentence contains a long clause that makes the sentence difficult to follow. Suggested edit: "We find that IGOS flights will likely provide a strong constraint on regional CO2 flux totals."

Fig 4 caption: Some readers may not be familiar with Taylor diagrams. You could include once sentence that explains which corners of the Taylor diagram are "better" and which are "worse." (I had to stare at the diagram for a minute to figure out that orange dots in the lower right are a better fit than those in other regions of the diagram.)

---

## Referee Comment (RC2) · Anonymous Referee #1 · 11 Oct 2016

General comments.

The paper presents an observing system simulation study investigating the information content of the planned IAGOS aircraft observations of atmospheric CO2 profiles, and projected impact of IAGOS observations on reducing uncertainty of surface CO2 flux estimates by inverse modeling. Paper is written well and useful, and can be published after revision. A minor revision is needed to address few weaknesses to the study and its presentation.

Detailed comments.

[Figure]

Page 2-Line 37 It would be useful to note that Niwa et al (2012) and Patra et al (2011) relied mostly on free-tropospheric part of the profiles. There are some earlier studies looking at aircraft vertical profiles and their use in inversions. Gloor et al (2000) have considered aircraft vertical profiles in their studies on observing network extension. A difference between fluxes estimated using near-surface observations and column average of the vertical profiles was discussed by Nakatsuka and Maksyutov (2009).

P3-L37 Equation 1 is written or described incorrectly; in place of Cini should be the result of forward simulation with initial concentration Cini.

P4-25 Equations 4b and 6 give impression that prior flux error covariance matrix is omitted. These equations look different from Jena inversion system described by Rodenbeck et al (2005). Authors should review the Equations (3-6, 14) in (Rodenbeck et al 2005) and explain the changes, in case there are some.

P10-L35 More discussion can be added on this topic. The transport model used in this study may not be best one for actually analyzing the IAGOS observations in PBL, due to a need to resolve plumes of anthropogenic CO2 transported from large cities near the airports. A relatively high model-observation mismatch of 5 ppm at 1 km as shown on Fig. 1 was found for CONTRAIL data. High model data mismatch (mdm) could partly be a result of applying low resolution (with respect to city plume size) model and meteorology, thus it should be considered as upper bound on mdm. Using the data uncertainty based on CONTRAIL mismatch for IAGOS looks justifiable with current transport model, and large data uncertainty may have resulted in relatively low flux uncertainty reductions in the order of 10

The ability of the low resolution model to simulate CO2 concentration in the megacity plumes is questionable, with possible underestimation of fossil CO2 component due to low model resolution (model is low biased), affecting the estimated fluxes.

Technical corrections.

P1-L14 Suggest correcting "ground- based" to "ground-based"

P1-L19 Suggest correcting "under constrained" to "underconstrained"

P2-L2 In "unevenly distributed observation network of observation can result" – sounds like "network of observation sites" would fit better.

P2-L16 Suggest correcting "Checa- Garcia" to "Checa-Garcia"

References

Gloor, M., S.-M. Fan, S. Pacala, and J. Sarmiento, Optimal sampling of the atmosphere for purpose of inverse modeling: A model study, Global Biogeochem. Cycles, 14(1), 407–428, doi:10.1029/1999GB900052, 2000.

Nakatsuka, Y. and Maksyutov, S.: Optimization of the seasonal cycles of simulated CO2 flux by fitting simulated atmospheric CO2 to observed vertical profiles, Biogeosciences, 6, 2733-2741, doi:10.5194/bg-6-2733-2009, 2009.

---

## Author Comment (AC1) · 6 Jan 2017

**Authors' responses to reviewers' comments:**

We would like to thank both the referees for their careful reviewing and constructive comments and suggestions for this manuscript. Our responses to the comments are as follows:

**[RC]:** Reviewer's comment **[AR]:** Authors' response **[ME]:** Manuscript edits & modification

**Reviewer # 1**

[RC] P2L37: It would be useful to note that Niwa et al (2012) and Patra et al (2011) relied mostly on free-tropospheric part of the profiles. There are some earlier studies looking at aircraft vertical profiles and their use in inversions. Gloor et al (2000) have considered aircraft vertical profiles in their studies on observing network extension.
A difference between fluxes estimated using near-surface observations and column average of the vertical profiles was discussed by Nakatsuka and Maksyutov (2009).

[AR]: These two references will be added to the manuscript and the text will be modified as follows:

> [ME]: "Both studies focused specifically on the estimation of the tropical terrestrial fluxes using mostly the free tropospheric part of the aircraft profiles. Gloor et al (2000) used aircraft vertical profiles in their studies for observing network extension. A difference between fluxes estimated using near-surface observations and column average of the aircraft vertical profiles was discussed by Nakatsuka and Maksyutov (2009). However, so far, the suitability of aircraft vertical profiles and their treatment when using them into inversions, given the transport modelling errors related to vertical mixing has not been addressed."

[RC] P3L37: Equation 1 is written or described incorrectly; in place of Cini should be the result of forward simulation with initial concentration Cini.

[AR]: $C_{ini}$ is the initial atmospheric mixing ratio of the transport model at the beginning of the simulation period.

[RC] P4L25: Equations 4b and 6 give impression that prior flux error covariance matrix is omitted. These equations look different from Jena inversion system described by Rodenbeck et al (2005). Authors should review the Equations (3-6, 14) in (Rodenbeck et al 2005) and explain the changes, in case there are some.

[AR]: This part of the text will be condensed and the equations have been modified to resemble those in Roedenbeck et al. 2005.

[RC] P10L35: More discussion can be added on this topic. The transport model used in this study may not be best one for actually analyzing the IAGOS observations in PBL, due to a need to resolve plumes of anthropogenic CO2 transported from large cities near the airports. A relatively high model-observation mismatch of 5 ppm at 1 km as shown on Fig. 1 was found for CONTRAIL data. High model data mismatch (mdm) could partly be a result of applying low resolution (with respect to city plume size) model and meteorology, thus it should be considered as upper bound on mdm. Using the data uncertainty based on CONTRAIL mismatch for IAGOS looks justifiable with current transport model, and large data uncertainty may have resulted in relatively low flux uncertainty reductions in the order of 10. The ability of the low resolution model to simulate CO2 concentration in the megacity
plumes is questionable, with possible underestimation of fossil CO2 component due to low model resolution (model is low biased), affecting the estimated fluxes.

[AR]: The discussion related to the transport model resolution will be added in the text.
> [ME]: The text will be modified as follows:
> "We must bear in mind that since the MOZAIC/IAGOS aircraft profiles are measured near the airports, which form areas of high anthropogenic emissions, it is likely that these observations are not truly representative of large areas. This fact has been taken into account, in this study, in a conservative way by estimating the model data mismatch uncertainty using the difference between $CO_2$ profiles from the CONTRAIL project and reanalysed TM3 fields (Sect. 2.2). However, better approaches for addressing this question of representativeness of aircraft profiles exist, for example, those described by Boschetti et al. 2015. A relatively high model-observation mismatch of 5 ppm at 1 km (as shown in Fig. 1) for CONTRAIL data could partly be a result of applying low-resolution (with respect to

plume size of anthropogenic $CO_2$ transported from large cities near the airports) model and meteorology and thus should be considered as an upper bound on the model data mismatch."

[RC] P1L14: Suggest correcting "ground- based" to "ground-based"
[AR]: Done

[RC] P1L19: Suggest correcting "under constrained" to "underconstrained"
[AR]: Done

[RC] P2L2: In "unevenly distributed observation network of observation can result" – sounds like "network of observation sites" would fit better.

> [ME]: Text will be modified as follows:
> "Lack of measurements in the atmosphere or an unevenly distributed network of observation sites can result in a poorly constrained regional carbon budget (Gurney et al. 2002)."

[RC] P2L16: Suggest correcting "Checa- Garcia" to "Checa-Garcia"

[AR]: Done

**Reviewer # 2**

[RC] P1L14 : I would include a noun after a word like "this" or "these" so it is clear to the reader what you are referring to in the sentence.
[RC] P1L14: consider changing "the benefit" to "a benefit"[AR]: The text will be modified to clarify this point:
> [ME]: "This finding highlights a benefit of utilizing atmospheric observations made onboard aircraft over surface measurements for flux estimation using inverse methods."

[RC] P1L20 : Suggested rewording: "with a reduction of posterior flux uncertainty of about 7 to 10%."
[AR]: Text will be modified as per the suggested rewording
> [ME]: "Our results show that the regions tropical Africa and temperate Eurasia, that are under-constrained by the existing surface based network, will benefit the most from these measurements, with the reduction of posterior flux uncertainty of about 7 to 10 %."

[RC] P1L22: add a comma before "and"
[AR]: Done
> [ME]: "Reliable prediction of climate change scenarios requires a thorough understanding the carbon-climate feedbacks in the earth system, and accurately estimating current sources and sinks of carbon is of prime importance."

[RC]: P1L24: What does "these" refer to here? I would include a noun after "these".
[AR]: The phrase "sources and sinks" will be added to this sentence.
> [ME]: While it is impossible to measure these sources and sinks directly everywhere around the globe, we may estimate these using the 'top-down' approach employing atmospheric observations in combination with knowledge of atmospheric transport and prior knowledge of the fluxes by inverse modelling.

[RC]: P1L28-29: Consider adding references to this sentence.
[AR]: The reference Gerbig et al.,2003 will be added to this sentence
> [ME]: Unfortunately, the estimates of surface fluxes using this approach are prone to large uncertainties that can largely be attributed to imperfections in the transport models and insufficient data coverage by the observation network (Gerbig et al., 2003).

[RC]: P1L35: Consider adding references to this sentence.
[AR]: references Stephens et al., 2007; Gerbig et al., 2008 will be added to this sentence
> [ME]: One of the dominant sources of transport model uncertainty is the inaccurate representation of the vertical mixing near the surface of the earth and hence the boundary layer height (Stephens et al., 2007; Gerbig et al., 2008).

[RC] P2L11-12: This point is true of short towers but not necessarily of tall towers
(e.g., the NOAA tall tower network in the US).

[AR]: The text will be modified to clarify this statement

[ME]: In addition, these measurements except those obtained from tall towers, are often not representative of large areas and provide information only at the local scale (Haszpra et al. 1999).

[RC] P2L14: Instead of "constraints", I would use a word like "limitations" ("... they have their limitations, too, which restricts their use for accurate flux ...."). The word "constraints" makes it sound like the satellite is constraining something (e.g., fluxes), not that the satellite has limitations.

[AR]: As per the suggestion, the word "constraints" will be replaced by "limitations"

[ME]: "However, they have their limitations, too, which limits their use for accurate flux estimation using inverse methods."

[RC] P2L41: Suggested rewording: "... investigate theoretical impacts of transport model ...."

[AR]: The text will be modified according to the suggested rewording

[ME]: "In this paper we employ synthetic data to investigate theoretical impacts of transport model uncertainties associated with boundary layer height on the fluxes retrieved by using passenger aircraft profiles in an inverse modelling set-up."

[RC] P3L10-17: Some of the information here seems redundant with information in the previous two paragraphs. You may want to eliminate these lines or condense with the previous two paragraphs.

[AR]: The text in these lines will be removed to avoid repetition with information in the previous paragraph.

[RC] P3L20: I prefer active voice (E.g., "section 3 presents results") over passive voice ("In section 2, the results are presented"). This wording could be a matter of personal preference.

[AR]: Active voice will replace passive voice in this sentence.

[ME]: "Section 2 describes the methods used that include estimation of the model representation error (Section 2.1), description of the inversion scheme (Section 2.2) and the experimental set-up (Section 2.3). Section 3 presents the results from the simulations and the conclusions are discussed in Section 4."

[RC] P3L29: A 4x5 degree resolution seems really coarse for $CO_2$ simulations. At this juncture, I imagine it would be difficult to find a higher resolution model and re-run all of the CO2 model simulations. However, somewhere in the paper (e.g., in a supplement), it might be useful to explain why you used this resolution and how the resolution affects your interpretation of the results.

[AR]: We agree that the resolution of the model used is not the best that is currently available. However, similar resolutions have been used in the past in other intercomparison projects like RECCAP (Peylin et al.,2013) and Houweling et al., 2015). The impact of the resolution is likely to be on the model-data mismatch calculated here using the CONTRAIL data, which should be taken as an upper bound. This fact will be incorporated in the discussion section.

[RC] P3L31: I would include a noun after "following."

[ME]: Text will be modified as follows:
"In the following paragraphs, we provide a brief description of the inversion system described in more detail in Roedenbeck et al. (2005)."

[RC] P4Eq. 4b: Many inverse modeling studies include an a priori covariance matrix that defines uncertainties in the prior and spatial/temporal covariances in these covariances. Instead, you have used a weighting factor (mu). It could be helpful to explain why you have chosen the latter approach over the former.

[AR]: The factor 'mu', in this manuscript, is not related to the prior flux uncertainty or the spatial and temporal covariances in the fluxes. It is simply a factor the scales the contribution of the prior flux constraint on the inversion. It is a ratio between the a-priori information and data constraint.

[RC]: P4L12: The symbol mu usually refers to the mean in statistics. A different variable name here could prevent confusion. Section 2.1.1: This section contains a lot of information about the fundamentals of Bayesian inverse modeling, and it appears that a lot of this information has already been published elsewhere. You could either condense this section or move the material to a supplement.

[AR]: The symbol mu cannot be changed due to the fact that it is a part of published work (Roedenbeck at al., 2005) and therefore will have to be used as such. As per the suggestion, this section will be condensed to avoid repetition with the published literature.

[RC] P4L38: Are there many stations that have weekly flasks? I know of a number of stations with daily flasks. You could change this sentence to "... made once per day or once per week ...."

[AR]: While most of the flask stations used in this study are weekly, there are about a couple that make measurements once a week. The text in this sentence will be modified as follows.

[ME]: "While stations based on flask observations have measurements made once per day or once per week, there also exist a growing number of continuously measuring stations with data provided typically half hourly or hourly."

[RC] P5L5-24: You could condense this information or move it to a supplement if it has been published elsewhere.

[ME]: This part of the manuscript will be condensed as follows:
"For surface network sites, to avoid a higher impact of the more frequent continuous observations compared to the less frequent flask observations, the data density weighting considers, for every observation, the number of observations $N_{surf}$ within the same week. The total uncertainty for that observation increases by a factor of $\sqrt{N_{surf}}$. These $N_{surf}$ measurements have their errors correlated and this error inflation by a factor of $\sqrt{N_{surf}}$ helps lessen the impact of measurements that are not independent of each other and hence their contribution to the cost function.

The aircraft is a moving platform, which means that the aircraft profiles span a considerable horizontal and vertical distance while making measurements. Therefore, in contrast to a fixed station, the $CO_2$ concentration along the profile can be expected to de-correlate due to distance, even if taken within a short period of time. We need to incorporate this fact in the de-weighting scheme."

[RC] P5L 37-38: This definition of model-data mismatch is confusing. For example, what simulated spatial averages are you referring to here? Does this definition include or exclude temporal averaging? Also, I think this definition of model-data mismatch is different than the definition used in other inverse modeling papers (e.g., Michalak et al., 2004, doi:10.1029/2003JD004422).

[AR]: This definition of model-data mismatch has been incorporated from the paper by Engelen et al., (2002) in which they have referred to this quantity as "External Representation Error". The term "model simulated spatial averages" refers to the spatial average over model grid box. This definition includes both spatial and temporal averaging.

[RC] P8L7: I would include a comma after "redistribution."
[AR]: Done

[RC] P8L19: What does "this" refer to? I would add a noun after "this."
[ME]: The text will be modified as follows:
"This similarity in sensitivity of posterior flux between simulation types C and S shows that the effect of the surface network dominates the flux retrieval from the observations using the combined network and indicates that the surface network stations largely contribute to the sensitivity of the retrieved flux to the uncertainty of the boundary layer height."

[RC] P8L28: The word "constraint" is potentially confusing here. You could use a different phrase like "evaluate the utility of the aircraft measurements ...."
[ME]: The text will be modified as follows:
"In this section, we evaluate the utility of aircraft measurements of $CO_2$ from IAGOS for constraining the regional carbon budget."

[RC] P8L39: The phrase "cause a constraint" may not be the optimal wording here. Instead, you could use a phrase like the following: "West winds mean that observations in these regions are sensitive to boreal fluxes."
[ME]: The text will be modified as follows:
"For instance, the value of the uncertainty reduction over North American boreal regions (75 %) is high inspite of insufficient surface stations in that region. This can be attributed to the impact of the westerly winds flowing over Temperate North America. West winds mean that observations in these regions are sensitive to boreal fluxes."

[RC] P9L8: I would re-phrase "least constraint in the fluxes." Instead, you could try "least constrained by the combined network ...".
[ME]: The text will be modified as follows:
"Tropical Asia is the least constrained by the combined network since it is not adequately covered by either of the networks- surface or the passenger aircraft."

[RC] P9L12: Throughout the article, the word "constraint" is used in a variety of different contexts with different meanings. I would use this word in a single context and choose different words in different contexts. In this line, I think the phrase "uncertainty reduction" might be more appropriate.
[ME]: The text will be modified as follows:

"Tropical and Eurasian temperate regions show the greatest change in the uncertainty reduction of the posterior fluxes on addition of pseudo observations from IAGOS (about 7 to 10 %)."

[RC]: P9L24: I would remove the word "however."
[ME]: Done
"The Tropics, on the other hand, show a comparable trend and increase in the change of flux uncertainty with up to 10 times fewer measurements than in the Northern hemisphere."

[RC] P9L25-26: I think this sentence is a run-on. I would put a period after "network" and start a new sentence with the word "hence." Also, I would add a noun after the word "this".
[ME]: Done
"This difference in uncertainty reduction change in the two regions is likely to be due to the fact that unlike the Northern hemisphere, the tropics are not well constrained by the existing network. Hence, the addition of IAGOS profiles leads to considerable constraint on the surface fluxes."

[RC] P10L7 and 10: What do the words "this" refer to in each sentence? I would add a noun after the word "this" in each case.
[ME]: The text will be modified as follows:
"In other words, this mismatch shows that the transport model uncertainties related to boundary layer height are very likely to be translated to the posterior flux when surface measurements are used as constraint in the inversion while these errors are not propagated to the retrieved flux when the aircraft profiles are used. This difference in the response of the flux retrieved using the two observation networks is likely to be due to the fact that vertical transport, whose effect we simulate by the redistribution of the tracer mass in the model profile at the location of the airports and surface stations, only redistributes the tracer mass between the boundary layer height and the free tropospheric part keeping the total tracer mass constant."

[RC] P10L22: I would change "reduce" to "decline".
[ME]: Done
"Although we only account for errors in fluxes due to vertical mixing in our simulations, we can say that flux estimation using aircraft profiles is expected to be more robust when aircraft profiles are used as constraint since the contribution of the boundary layer height uncertainty to the overall transport model error is likely to decline."

[RC] P10L28: This sentence contains a long clause that makes the sentence difficult to follow. Suggested edit: "We find that IGOS flights will likely provide a strong constraint on regional CO2 flux totals."

[ME]: The text will be modified as follows:
"Furthermore, on estimating the impact that the $CO_2$ measurements made onboard the IAGOS fleet are likely to have on the regional carbon budget once they are available, we find that the IAGOS flights will likely provide a strong constraint on regional flux totals."

[RC]: Fig 4 caption: Some readers may not be familiar with Taylor diagrams. You could include once sentence that explains which corners of the Taylor diagram are "better" and which are "worse." (I had to stare at the diagram for a minute to figure out that orange dots in the lower right are a better fit than those in other regions of the diagram.)

[ME]: The figure caption will be modified as follows:
"Figure 4: Taylor diagram showing the correlation coefficient, standard deviation and root mean square difference of the concatenated time series of the monthly posterior fluxes from the TransCom3 land regions. Standard deviation of the time series is depicted on the vertical axis while the correlation coefficient with respect to the true flux time series is shown on the circular arc of the diagram. Root mean square difference of the time series is shown on the green arcs. Points S, A, C represent the simulations using measurements from only the surface stations, only the aircraft profiles and the combined network (Surface + Aircraft) respectively. 'a' denotes the control case simulation with well known boundary layer height while 'b' denotes simulations using reshuffled profiles with "wrong" boundary layer height. Points closer to the True Flux point, near the lower right corner of the diagram are a better fit."